# Classical Simulation of Quantum Circuits Using Reinforcement Learning: Parallel Environments and Benchmark

**Xiao-Yang Liu**[1,2] **and Zeliang Zhang**[2]*
[1]Rensselaer Polytechnic Institute, [2]Columbia University,
`liux33@rpi.edu, XL2427@columbia.edu, hust0426@gmail.com`

## Abstract

Google's "quantum supremacy" announcement [3] has received broad questions from academia and industry due to the debatable estimate of $10,000$ years' running time for the classical simulation task on the Summit supercomputer. *Has "quantum supremacy" already come? Or will it come in one or two decades later?* To avoid hasty advertisements of "quantum supremacy" by tech giants or quantum startups and eliminate the cost of dedicating a team to the classical simulation task, we advocate an open-source approach to maintain a trustable benchmark performance. In this paper, we take a reinforcement learning approach for the classical simulation of quantum circuits and demonstrate its great potential by reporting an estimated simulation time of less than $4$ days, a speedup of $5.40\times$ over the state-of-the-art method. Specifically, we formulate the classical simulation task as a *tensor network contraction ordering* problem using the K-spin Ising model and employ a novel Hamiltonian-based reinforcement learning algorithm. Then, we evaluate the performance of classical simulation of quantum circuits. We develop a dozen of massively parallel environments to simulate quantum circuits. We open-source our parallel gym environments and benchmarks.

## 1   Introduction

Google proudly announced "achieving quantum supremacy" [3] with its 53-qubit Sycamore circuits back in 2019, which was later challenged by researchers claiming to have pulled ahead of Google on classical computers. *Quantum supremacy* [46] aims to demonstrate that a programmable quantum device can solve a problem that no classical computer can solve in any feasible amount of time, irrespective of the usefulness of the problem. As illustrated in Fig. 1, for the problem of random number generation, Google's "quantum supremacy" announcement [3] relied on an estimated simulation time of $10,000$ years on the Summit supercomputer, while existing works reduce it to less than $21$ days and scale the number of quantum qubits up to around $100$. This raises the debate: ***Has "quantum supremacy" already come? Or will it come in one or two decades later?*** Our goal is to use machine learning methods to derive the best performance curves for the classical simulation task, to ***settle down the present debate and suggest***

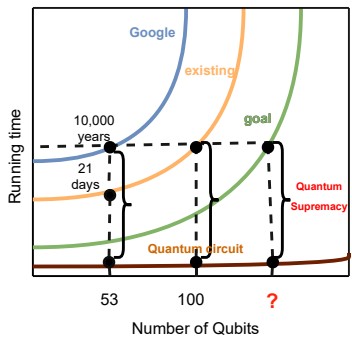

Figure 1: Running time of different quantum circuit simulation methods.

---

*Zeliang Zhang completed this work as a research assistant at Columbia University.

37th Conference on Neural Information Processing Systems (NeurIPS 2023) Track on Datasets and Benchmarks.

*that achieving "empirical quantum supremacy" requires continuing quantum hardware developments without an unequivocal first demonstration*.

There are several existing works on the classical simulation of quantum circuits [8]. One promising method is using tensor networks [42] since quantum circuits can be naturally represented as tensor networks, i.e., executing a quantum gate is mathematically a tensor contraction operation. Therefore, establishing a benchmark curve for the classical simulation task is mathematically searching for the optimal tensor network contraction ordering (TNCO), which is a combinatorial optimization problem [25], a variant of the well-known traveling salesman problem (TSP) where reinforcement learning (RL) algorithms [33] have shown powerful capability. *We are motivated to take a reinforcement learning approach to establish state-of-the-art performance.*

There is a debate on the estimate of running time for classical simulation tasks, as illustrated in Fig. 2. Huang *et al.* [21] used a heuristic tensor network approach on a computing cluster and estimated the simulation time to be 21 days. Meirom *et al.* [34] used a reinforcement learning algorithm where the policy network used a graph neural network and reported an order-wise reduction compared to the best heuristic method. However, there is no available dataset for training and benchmarking machine learning algorithms, and it is reasonable to maintain a publically trustable performance curve for the coming quantum supremacy.

To avoid hasty advertisements of "quantum supremacy" by tech giants or quantum startups and eliminate their cost of dedicating a team to the classical simulation of quantum circuits, establishing a standard benchmark is important. In this paper, we take a reinforcement learning (RL) approach for the classical simulation of quantum circuits and demonstrate its great potential by reporting an estimated simulation time of less than 4 days, a speedup of $5.40\times$ over the state-of-the-art method. The result demonstrates that *the "quantum supremacy" claim still lacks an unequivocal first demonstration*. Specifically, we adopt the K-spin Ising model for the classical simulation of quantum circuits, i.e., the tensor network contraction order (TNCO) problem, and employ the Hamiltonian-based reinforcement learning algorithm to minimize the number of multiplications. Then, we establish standard criteria to evaluate the simulation performance for Google's Sycamore circuits. We develop a dozen of massively parallel environments for training and evaluating RL agents. We release multiple datasets, including tensor-train, synthetic, and sycamore quantum circuits, and benchmark curves, including OPT-Einsum, Cotengra, and our RL method, on Github at `https://github.com/XiaoYangLiu-FinRL/RL4QuantumCircuits`.

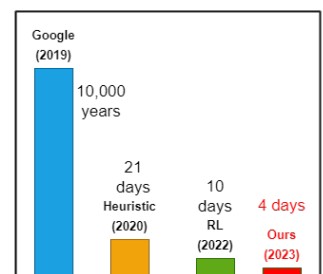

Figure 2: Debatable time estimates of classical simulation of Google's Sycamore circuits.

We hope the AI/ML community and quantum physics community will collaborate to maintain reference curves for validating an unequivocal first demonstration of "empirical quantum supremacy".

## 2 Related Works

**Random circuit sampling**: Random Circuit Sampling (RCS) [35] is an approach that has garnered considerable attention within the quantum computing community. It involves the generation of random quantum circuits, which are subsequently subjected to measurement procedures. By sampling the measurement outcomes, RCS enables the evaluation of the distribution of classical output probabilities, providing valuable insights into the computational power of noisy intermediate-scale quantum devices. While many works [27, 38, 35] demonstrate the hardness of RCS, highlighting its computational complexity, the reinforcement learning-based method [34] presents a promising way to accelerate the classical simulation of RCS.

**Quantum circuits for random number generation task**: One notable example of the application of quantum circiuits [53] is the random number generation. It is NP-hard since simulating a probability distribution on classical computers involves an exponential time complexity. For the problem of random number generation, Google's Sycamore circuit [3] was a milestone in quantum computing, which used the random circuit sampling technique to achieve the random selection process and claimed to demonstrate "quantum supremacy". However, it is hard for most researchers to access

the computing resources of quantum circuits for study. It is critical to simulate the quantum circuits and establish the dataset and benchmark performance to verify such scientific claims.

**Classical simulation of quantum circuits**: In the field of classical simulation, several approaches have been proposed to simulate quantum circuits, including the matrix representation methods [8], variational algorithms [51], and tensor network methods [42]. Among these, the tensor network method leverages the mathematical framework of tensor networks to approximate quantum states and perform efficient simulations. Tensor network methods, such as the matrix product state (MPS) [16] and projected entangled pair state (PEPS) [59], have shown promising results in simulating both one-dimensional and two-dimensional quantum systems. In this paper, we take the tensor network method for simulating quantum circuits.

**Tensor network contraction** [48] and the search for the optimal contraction path [4] have garnered significant attention in computational physics and quantum information theory. Various techniques [9] are employed to identify the most efficient way to contract tensors, minimizing computation cost. These advancements enable efficient simulation of quantum circuits. There are two popular open-source libraries, OPT-Einsum [13] and Cotengra [18] (CTG-Greedy and CTG-Kahypar), while Cotengra achieves state-of-the-art performance for most tensor networks.

**Ising model**: The Ising model [11] can serve as a unified formulation of combinatorial optimization problems, including graph coloring, maximum cut problems, and traveling salesman problems. The Ising model describes a system of interacting spins that can be either up or down, representing binary variables in studied problems. By mapping the studied problem onto an Ising model, the problem can be reformulated as finding the configuration of spins that minimizes the corresponding energy function [57]. The tensor network contraction problem can be analyzed into one kind of combination problem, especially the TSP. Thus, it motivates us to model the tensor network contraction problem as the Ising model and learn the optimal tensor contraction path by minimizing the energy function to reduce the computation complexity.

**AI/ML + X**: In recent years, the integration of artificial intelligence (AI) and machine learning (ML) techniques in scientific research has revolutionized various domains [12, 26, 6, 23]. The application of AI/ML in scientific fields, often referred to as AI+X or machine learning for science, has brought about significant advancements and novel approaches to solving complex problems [54]. For example, tensor network factorization can be used for probabilistic modeling [15]. The quantum entanglement can be built as a fundamental connection with deep neural network design [28]. Quantum K-spin Hamiltonian Regularization is proposed to stabilize the reinforcement learning process [29]. A recent study uses the tensor network-based quantum circuits for image classification [19].

## 3 Classical Simulation of Quantum Circuits Using Tensor Networks

We use uppercase calligraphic letters to denote tensors, e.g., $\mathcal{X} \in \mathbb{R}^{I \times J \times K}$, uppercase and lowercase boldface letters to denote matrices and vectors, e.g., $\boldsymbol{X} \in \mathbb{R}^{I \times J}$ and $\boldsymbol{x} \in \mathbb{R}^{I}$.

### 3.1 Quantum Circuits

**Qubits**: A qubit in a superposition state can be represented as $|\phi\rangle = \alpha_0 |0\rangle + \alpha_1 |1\rangle$, where $\alpha_0, \alpha_1 \in \mathbb{C}$, and $|\alpha_0|^2 + |\alpha_1|^2 = 1$. For $n$ qubits, we use a linear combination of $2^n$ coefficients and states $|0...0\rangle, |0...1\rangle, \cdots |1...1\rangle$, respectively,

$$|\psi\rangle = \alpha_{0...0} |0...0\rangle + ... + \alpha_{1...1} |1...1\rangle , \tag{1}$$

where $\alpha_{0...0}, ..., \alpha_{1...1} \in \mathbb{C}$, and $|\alpha_{0...0}|^2 + ... + |\alpha_{1...1}|^2 = 1$.

**Quantum gates**: Single- and double-qubit quantum gates are building blocks of quantum circuits,

- Single-qubit gate:

$$\sqrt{\boldsymbol{X}} = \frac{1}{\sqrt{2}} \begin{bmatrix} 1 & -i \\ -i & 1 \end{bmatrix}, \ \ \sqrt{\boldsymbol{Y}} = \frac{1}{\sqrt{2}} \begin{bmatrix} 1 & -1 \\ 1 & 1 \end{bmatrix}, \ \ \sqrt{\boldsymbol{W}} = \frac{1}{\sqrt{2}} \begin{bmatrix} 1 & -\sqrt{-i} \\ \sqrt{-i} & 1 \end{bmatrix}, \tag{2}$$

- Double-qubit gate:

$$\text{fSim}(\theta, \phi) = \begin{bmatrix} 1 & 0 & 0 & 0 \\ 0 & \cos\theta & -i\sin\theta & 0 \\ 0 & -i\sin\theta & \cos\theta & 0 \\ 0 & 0 & 0 & \exp^{-i\phi} \end{bmatrix}, \tag{3}$$

where $\theta \approx \pi/2$ and $\phi \approx \pi/6$ are used in the Sycamore quantum circuits.

**Quantum circuits** consists of a sequence of quantum gates. For a given initial state $|\psi_0\rangle$ and a quantum circuit $\boldsymbol{U} = \boldsymbol{U}_m \cdots \boldsymbol{U}_1$, the final state is $|\psi\rangle = \boldsymbol{U}_m \times \cdots \times \boldsymbol{U}_1 |\psi_0\rangle$. Fig. 3 illustrates an example of $4$ qubits and a circuit $\boldsymbol{U}$ of $m = 2$ cycles. the initial state $|\psi_0\rangle = |0000\rangle$. The quantum circuit $\boldsymbol{U}$ takes an initial state $|\psi_0\rangle$ of $n$ qubits as input, performs $m$ cycles of gate operations, and outputs a bit-string $i_1...i_n$ of length $n$.

In the $i$-th cycle, two operations are performed:

1. A single-qubit gate $\boldsymbol{R}_i^j \in \mathbb{C}^{2\times 2}$ randomly selected from set $\{\sqrt{\boldsymbol{X}}, \sqrt{\boldsymbol{Y}}, \sqrt{\boldsymbol{Z}}\}$ is applied to $|\psi_{i-1}\rangle_j$, resulting in state $|\psi_i\rangle_j = \boldsymbol{R}_i |\psi_{i-1}\rangle_j$.

2. Then, execute a two-qubit quantum gate $\boldsymbol{U}_i^p \in \mathbb{C}^{4\times 4}$ to $|\psi_{i-1}\rangle_p$ and $|\psi_{i-1}\rangle_{p+1}$ and obtain a new state $|\psi_i\rangle_{p,p+1}$. Specifically, during an odd cycle, qubit pairs with indices originating at $0$, such as $(0,1),(2,3)$, are chosen. For even cycles, qubit pairs with indices starting at $1$ are selected, including pairs such as $(1,2),(3,4)$. This systematic approach ensures the appropriate application of the $\boldsymbol{U}_i^p$ operation on the designated qubit pairs.

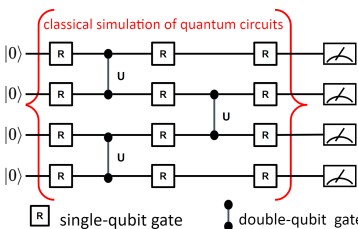

Figure 3: A quantum circuit where the classical simulation lies in the middle.

Before measurement, each qubit $|\psi_m\rangle_j$ undergoes a random single-qubit quantum gate $\boldsymbol{R}_{m+1}^j$. The random binary bit-string $i_1...i_n$ is sampled from the probability distribution $P = |\langle i_1 i_2...i_n | |\psi_{m+1}\rangle|^2$, which is obtained from the measurement outcome. More details about the random circuit sampling can be found in Appendix A.

## 3.2 Random Circuit Sampling

**Quantum supermacy**: Quantum supremacy refers to a major milestone in quantum computing, representing the point at which a quantum computer can solve a specific problem that is practically infeasible for classical computers to solve within a reasonable timeframe. It is a demonstration of the superior computational power of quantum systems compared to classical counterparts. Achieving quantum supremacy signifies the ability of a quantum computer to perform computations exponentially faster than even the most powerful classical supercomputers.

**A quantum circuit defines a distribution**: Quantum circuit is a sequence of quantum operations or gates applied to a set of qubits. These gates manipulate the quantum state of the qubits, transforming them according to the specific operations performed. At the end of a quantum circuit, measurements are typically performed on the qubits, extracting classical information from the quantum system. The results of these measurements are probabilistic, meaning that they occur with certain probabilities. Therefore, a quantum circuit defines a distribution by specifying how these probabilities are distributed among the different possible measurement outcomes.

For example, given a quantum circuit $U$, the initial state $|0\rangle$, and the output sampling bit-string $x = a_1 a_2...a_n \in \{0,1\}^n$, the modeled probability is $p(x) = |\langle x|U|0\rangle|^2$. The Google team used [1] the following noise model for the noisy samples produced by the Sycamore circuits,

$$N_c(x) = \phi P_C + (1-\phi)2^{-n}, \tag{4}$$

where $\phi$ is a fidelity parameter describing the quality of the sample, $P_C$ is the probability distribution defined by the quantum circuit. The parameter $\phi$ is estimated by follows,

$$\phi = \prod_{g\in\mathcal{G}_1}(1-e_{g_1})\prod_{g\in\mathcal{G}_2}(1-e_{g_2})\prod_{q\in\mathcal{Q}_1}(1-e_q), \tag{5}$$

where $\mathcal{G}_1$ is the set of single-qubit gate, $\mathcal{G}_2$ is the set of double-qubits gate, and $\mathcal{Q}_1$ is the set of qubits. Google set [1] $e_{g_1} = 0.16\%$, $e_{g_2} = 0.62\%$ and $e_q = 3.8\%$.

**XEB** [2]: XEB stands for "Linear Cross-Entropy Benchmarking" and is a metric used to assess the performance of quantum processors. It quantifies the fidelity of a quantum computer's output compared to a classical reference model. Mathematically, XEB is defined as:

$$\mathcal{F}_{XEB}(x) = 2^n \frac{1}{M} \sum_{x \in \mathcal{D}} p(x) - 1, \tag{6}$$

where we have a collection of $M$ random bit-strings $\mathcal{D} = \{x_1, ..., x_M\}$. If a random quantum circuit runs without errors, we have $\mathcal{F}_{XEB} = 1$. If bit-strings are sampled from a classical uniform distribution, we have $\mathcal{F}_{XEB} = 0$.

By evaluating the XEB metric, researchers can assess the performance of quantum circuits. It provides a quantitative measure of how well a quantum computer reproduces the expected outcomes and is a valuable tool for evaluating and improving quantum computing technologies.

### 3.3 Classical Simulation Using Tensor Networks

**The classical simulation task** aims to efficiently calculate $\boldsymbol{U} = \boldsymbol{U}_m \cdots \boldsymbol{U}_1$ in (7) on classical computers, as marked red in Fig. 3. It is mathematically a combinatorial optimization problem *tensor network contraction ordering (TNCO)*.

$$|\psi\rangle = \underbrace{\boldsymbol{U}_m \times \cdots \times \boldsymbol{U}_1}_{\text{classical simulation task}} |\psi_0\rangle. \tag{7}$$

**Tensor contraction operation**: Given two tensors $\mathcal{X} \in \mathbb{R}^{I \times J \times K}$ and $\mathcal{Y} \in \mathbb{R}^{K \times M \times N}$, their contraction results in a 4D tensor $\mathcal{Z} \in \mathbb{R}^{I \times J \times M \times N}$ where

$$\mathcal{Z}_{i,j,m,n} = \sum_{k=1}^{K} \mathcal{X}_{i,j,k} \mathcal{Y}_{k,m,n}, \tag{8}$$

which takes $IJKMN$ multiplications. Using the tensor diagram representation, as in Fig. 4, a node denotes a tensor and an edge denotes a tensor contraction operation.

**Tensor network representation**: Leaving the input quantum bits and the final measurement out, the quantum circuit $\boldsymbol{U} = \boldsymbol{U}_m \cdots \boldsymbol{U}_1$ can be represented as a tensor network. Specifically, a single-qubit gate $\boldsymbol{R}$ is represented as a matrix (a 2D tensor), while a double-qubit quantum gate $\boldsymbol{U}$ is represented as a 4D tensor. Using tensor diagrams, a quantum circuit in Fig. 3 is mapped into a tensor network in Fig. 4. Note that the final result of the classical simulation task is an 8D tensor shown in Fig. 4. We provide more details about the tensor representations in Appendix B.

**Simulation efficiency**: We would like to illustrate that the contraction ordering for calculating $\boldsymbol{U} = \boldsymbol{U}_m \cdots \boldsymbol{U}_1$ is critical to the simulation efficiency. Different orderings may result in a significant difference in the number of multiplications. The difference may be hundreds of orders for quantum circuits with a large number of qubits and many cycles [34]. For a simple example in Fig. 4, two contraction orderings (blue and red) for the same quantum circuit involve 976 and 5056 multiplications, respectively.

## 4 Tensor Network Contraction Ordering Using Reinforcement Learning

First, we formulate the classical simulation task as a combinatorial optimization problem, i.e., tensor network contraction order (TNCO). Then, we adopt a K-spin Ising model whose Hamiltonian is used as the loss function to train a policy network via curriculum learning. We also provide a pool of implementation tricks to improve training efficiency and boost performance.

### 4.1 Problem Formulation

**Tensor network contraction order (TNCO)**. Given a tensor network $G = (V, E)$, a contraction path $P = (e_1, \ldots, e_{n-1}), e_t \in E_t$, and a corresponding sequence of graphs $(G_1, \ldots G_{n-1})$, the

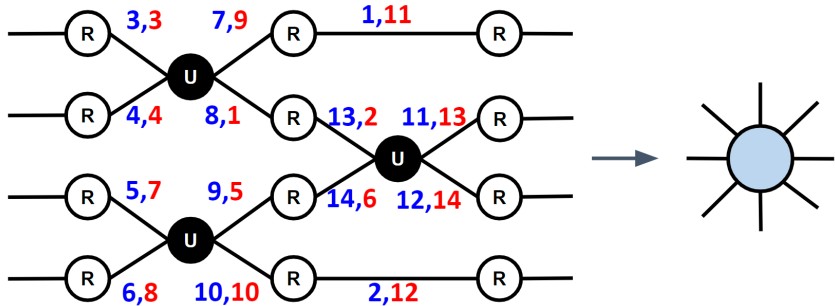

Figure 4: For the quantum cirtuis in Fig. 3, two contraction orderings (blue and red) involove 976 and 5056 multiplications, respectively.

goal is to find a path $P$ with minimum cost,

$$P^*(G) = \operatorname{argmin}_P \sum_{t=1}^{n-1} R_t(e_t) \tag{9}$$

$$\text{s.t. } P = (e_1, \ldots, e_{n-1}), e_t \in E_t,$$

where $E_t$ is the set of remaining edges between tensors, $G_t$ is the tensor network after $t$-th tensor contraction and the reward $R_t$ is defined as the number of multiplications for the tensor contraction along edge $e_t$. This formulation is consistent with [34].

Consider an example in Fig. 5, graph $G_1$ has $V_1 = (1, 2, 3, 4), E_1 = (k, j, m, i, s), w = \{K, J, M, I, S\}$. Assuming that the first contraction operation is on index $m$ in $G_1$, tensors 3 and 4 are contracted into tensor 34 at a computation cost of $IJMS$ multiplications. Then, the graph is updated to $G_2$ with $V_2 = (1, 2, 34), E_2 = (k, ij, s), w = \{K, IJ, S\}$. Assuming the second con-

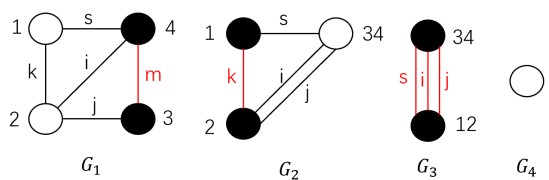

Figure 5: An example of tensor network contraction.

traction operation is on index $k$ in $G_2$, tensor 1 and 2 are contracted into tensor 12 at a computation cost of $SKIJ$ multiplications. The updated graph $G_3$ has $V_3 = (12, 34), E_3 = (sij), w = \{sij\}$. Finally, tensors 12 and 34 are contracted into a real number using $SIJ$ multiplications. The total number of multiplications is $IJMS + SKIJ + SIJ$.

### 4.2 The Proposed Reinforcement Learning Method

First, we reformulate (9) into an Ising model. Then, we extend it to a K-spin Ising model, which allows training a policy network using the curriculum learning method [5].

**TNCO using Ising model**: The Ising model of TNCO problem uses $N^2$ spins $\boldsymbol{x}_{u,j}$, where $u$ denotes the tensor and $j$ denotes its order in the TNCO path. We use $J_{u,v}^i$ to denote the cost introduced by the tensor contraction between $u$ and $v$ for the $i$-th order, *i.e.*, the number of multiplications. The energy of the original TNCO problem has three terms. The first term requires each tensor to appear at least once in the TNCO path. The second term requires there are exactly two tensors selected at order $j$ along a path. The third term measures the contraction cost at order $j$. These are encoded in the following Hamiltonian:

$$H(\boldsymbol{x}) = \sum_{i=1}^{N-1} \left\{ \left(2 - \sum_{u=1}^{N-i} \boldsymbol{x}_{u,i}\right)^2 + \sum_{u=1}^{N}\sum_{v=1}^{N} J_{u,v}^i \boldsymbol{x}_{u,i}\boldsymbol{x}_{v,i} \right\}. \tag{10}$$

As shown for example, in Fig. 6, it can be computed by $H(x) = (2 - \boldsymbol{x}_{1,1} - \boldsymbol{x}_{2,1} - \boldsymbol{x}_{3,1} - \boldsymbol{x}_{4,1})^2 + (2 - \boldsymbol{x}_{1,2} - \boldsymbol{x}_{2,2} - \boldsymbol{x}_{3,2})^2 + (2 - \boldsymbol{x}_{1,3} - \boldsymbol{x}_{2,3})^2 + w_1(1,4)\boldsymbol{x}_{1,1}\boldsymbol{x}_{4,1} + w_1(1,2)\boldsymbol{x}_{1,1}\boldsymbol{x}_{2,1} + w_1(2,3)\boldsymbol{x}_{2,1}\boldsymbol{x}_{3,1} + w_1(2,4)\boldsymbol{x}_{2,1}\boldsymbol{x}_{4,1} + w_1(3,4)\boldsymbol{x}_{3,1}\boldsymbol{x}_{4,1} + w_2(1,3)\boldsymbol{x}_{1,2}\boldsymbol{x}_{3,2} + w_2(1,2)\boldsymbol{x}_{1,2}\boldsymbol{x}_{2,2} + w_2(2,3)\boldsymbol{x}_{2,2}\boldsymbol{x}_{3,2} + w_3(1,2)\boldsymbol{x}_{1,3}\boldsymbol{x}_{2,3}$

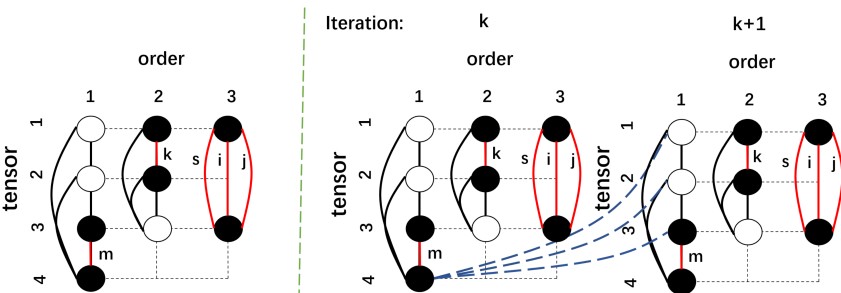

Figure 6: Illustration of TNCO problem: conventional Ising model vs. our K-spin Ising model.

**TNCO using K-spin Ising model**: Our solution takes K steps to solve the TNCO problem, i.e., $\boldsymbol{x}^1 \to \boldsymbol{x}^2 \to \cdots \to \boldsymbol{x}^K$, which can be encoded in the following Hamiltonian,

$$H(\boldsymbol{x}^1, \cdots, \boldsymbol{x}^K) = \sum_{k=1}^{K} H(\boldsymbol{x}^k) + \sum_{k=1}^{K} \sum_{i_1 \in V^1} \cdots \sum_{i_k \in V^k} J_{i \cdots i}^{1 \cdots k} \boldsymbol{x}_{i_1}^1 \cdots \boldsymbol{x}_{i_k}^k, \tag{11}$$

where $J_{i_1 \cdots i_k} = \sum_{k=2}^{K} \gamma^{K-k} w(i_1, i_2) w(i_2, i_3) \cdots w(i_{k-1}, i_k)$.

**Policy network**: We build the policy network with a transformer neural network. We represent the tensor network with an undirected graph, demonstrated as a symmetric matrix $\boldsymbol{M}$. If $i$-th and $j$-th tensors are connected with a shared index $d$, then $\boldsymbol{M}_{i,j} = d$. The policy network takes $\boldsymbol{M}$ as input and generates the contraction ordering $P = (e_1, ..., e_{n-1})$, where $e_t \in E_t$ is the connected edge between $u$-th and $i$-th tensors $\boldsymbol{x}_{u,i}$.

**Challenges**: Several challenges for using RL algorithms include the wide dynamic range, huge search space, slow convergence due to Heavy-tailed cost distribution, incorporating existing solvers, and credit assignment problem [34]. The sampling bottleneck is a major challenge in training RL agents [32]. Next, in Section 4.3, we develop a pool of implementation tricks to overcome these challenges, including massively parallel gym-environment, learning to optimize, dual replay buffers, swarm intelligence, and curriculum learning.

### 4.3 Implementation Tricks

**Massively parallel gym-environment**. The sampling bottleneck is a major challenge in training RL agents. We implement a massively parallel gym-environment to accelerate the sampling efficiency. We initialize $N$ independent environments and generate and forward different tensor network contraction paths to these environments at each step. Each environment performs the contraction operations according to the given path, yielding the next state and calculating the number of multiplications as rewards. We store these transitions, including the tensor networks and their corresponding multiplications, in the replay buffer.

**Learn to optimize**. We adopt the learn-to-optimize (L2O) strategy [10] for training an RL agent. Instead of using conventional optimizers like Adam or Nestrov, we employ a long-short-term memory (LSTM) network. It takes the current gradient and loss value as inputs and produces gradients. This L2O strategy effectively accelerates the convergence process, particularly in the presence of a Heavy-tailed cost distribution. This approach enables faster and more efficient training by dynamically adapting the parameter updates based on the current gradient and loss value.

**Dual replay buffers.** We maintain two replay buffers, one saves the tensor contraction ordering from the agent in an iterative rollout manner, and the other saves high-quality tensor contraction orders so far. The reason for two replay buffers is that we want to avoid high-quality tensor contraction ordering being deleted by the rule of first in, first out (FIFO) due to the GPU memory limit. We use the Hamiltonian value to measure the quality. We sample from two replay buffers alternatively during the training process, which stabilizes the RL training.

**Swarm intelligence.** We train multiple optimizers since different optimizer is easily stuck into local optima [7, 39]. To mitigate the impact, different optimizer shares knowledge with others regularly, thus escaping local optima and approaching the global optima. Specifically, during training,

optimizers keep the top-$N$ solutions while optimizers share them. Rather than directly using the orderings for training, the optimizer will manually add some noise to the shared orders to search for the solution in the neighborhood of that local optimum.

**Curriculum learning.** We apply curriculum learning method [5, 17, 62]. With an acceptable performance on small-scale tensor network contraction problems, the optimizer has successfully escaped multiple local optima. The optimizer may approach the global optima when varying the studied tensor networks from small to large scale. We have two tricks for measuring the problem scale: the number of iterations and parameter freezing strategies.

- The first trick is varying the number of iterations, $K$. A large $K$ may result in insufficient optimization, while in contrast, small-scale optimization problems corresponding to a small $K$ may experience over-fitting. Therefore, starting with an initial large $K$, we gradually decrease $K$ during the training process.

- The second trick is masking network parameters, i.e., freezing different ratios of parameters. Specifically, at first, we randomly freeze $90\%$ of the parameters and optimize the rest $10\%$ parameters, which is easy to optimize. Then, we gradually decrease the ratio of masked parameters. It can help the optimizer escape the local minima.

Note that the two curriculum learning tricks can be integrated with the swarm intelligence trick to boost performance.

## 5 Performance Evaluation and Benchmark

We present experimental results on both synthetic tensor networks and Google's Sycamore circuits. Further, we investigate the scalability performance of our method for large-scale quantum circuits that have a similar structure to Google's Sycamore circuits.

### 5.1 Baseline Methods

We provide the following five baselines for verification purposes:

- OE-GREEDY [13]: an open-source solver from OPT-Einsum library, which uses the greedy search algorithm to find the optimal tensor contraction path.

- CTG-Greedy [18]: an open-source solver from the Cotengra library, which uses the greedy search algorithm to find the optimal tensor contraction path.

- CTG-Kahypar [18]: A robust graph partitioning-based solver from the Cotengra library, which achieves state-of-the-art results in many tensor networks.

- ACQDP [21, 22]: Using the stem optimization, including the hypergraph partitioning, local optimization, and dynamic slicing, to search the optima tensor contraction order[2].

- RL-TNCO [34]: an RL approach combined with graph neural networks to search the optimal tensor contraction order, which is one of the state-of-the-art methods for the classical simulation of quantum circuits[3].

**Experiment setup**. We conducted all experiments on a DGX-2 server with NVIDIA A100 GPUs, each of which consists of 48 GB device memory. There are two Intel(R) Xeon(R) Gold 5118 CPUs. Each of CPUs has 12 cores @2.30GHz supporting 24 hardware threads. There are 128 GB DDR4 memories on the server. We set the learning rate $\eta$ as $3 \times 10^{-3}$ with a decay factor 0.9 (reduce the learning rate every $1,000$ epochs), and the total number of epochs is $1,000,000$. We set $K = 3$ for our K-spin Ising model. The number of parallel environments is 2048, while the large-scale cases use 1024. When the number of tensors increases, the GPU memory consumption increases; therefore, we use 1024 environments for large-scale tensor networks.

---

[2]It is not generalizable to all tensor networks. We only add ACQDP's performance in Sycamore circuits.

[3]The codes are not public, so we took results from Table 3 of the paper [34].

## 5.2 Verification on Synthetic Tensor Networks

**Synthetic tensor-train networks**. We tested tensor-train networks with tensors from $400$ up to $2000$. From Table 1, our RL-Ising method achieves a speedup of $2\times$ over the CTG-Kahypar method for tensor size from $400$ to $1000$. Both OE-Greedy and CTG-Greedy cannot work for the problem instances with over $1,500$ tensors, while the RL-Ising method exhibits good scalability, outperforming CTG-Kahypar by a speedup of $1.73\times$ for tensor sizes $1500$ and $2000$.

Table 1: Results on synthetic tensor-train networks.

| Tensors | 400 | 600 | 800 | 1000 | 1500 | 2000 |
|---|---|---|---|---|---|---|
| Scale | $\times 10^{120}$ | $\times 10^{180}$ | $\times 10^{241}$ | $\times 10^{301}$ | $\times 10^{451}$ | $\times 10^{602}$ |
| OE-Greedy [13] | 17.22 | 27.67 | 4.44 | 3.83 | - | - |
| CTG-Greedy [18] | 10.33 | 16.60 | 2.67 | 4.28 | - | - |
| CTG-Kahypar [18] | 10.23 | 16.60 | 4.67 | 4.26 | 14.12 | 4.57 |
| RL-Ising | **5.16** | **8.28** | **2.14** | **2.14** | **7.01** | **2.29** |

**Synthetic random tensor networks**. We generate random tensor networks with the same settings in [34]. Random networks were generated with varying tensors, namely 25, 50, 75, and 100. The connections in these networks were assigned an average degree of 3, with the degrees of individual nodes independently drawn from the set $\{2, 3, 4, 5, 6\}$. We report the mean and median results of 5 random instances for each size. As given in Table 2, our RL-Ising method outperforms all baselines with a speedup of $3.98\times$.

Table 2: Number of multiplications for synthetic random tensor networks.

| Qbits | 25 | 50 | 75 | 100 |
|---|---|---|---|---|
| Scale | $\times 10^4$ | $\times 10^7$ | $\times 10^{10}$ | $\times 10^{12}$ |
| OE-Greedy [13] | 53.7/27.7 | 75.4/11.3 | 104/4.5 | 5296/26.4 |
| CTG-Greedy [18] | 40.3/20.3 | 12.8/ 4.2 | 8.3/0.9 | 27.9/ 2.2 |
| CTG-Kahypar [18] | 46.4/24.8 | 13.4/ 4.3 | 4.1/0.4 | 54.2/ 1.2 |
| RL-TNCO [34] | 13.1/12.5 | 3.2/ 1.8 | 1.2/0.2 | 5.5/ 1.8 |
| RL-Ising | **12.5/11.4** | **2.5/ 1.5** | **0.7/0.1** | **4.9/ 1.1** |

## 5.3 Benchmark Performance on Google's Sycamore Circuits

Google's Sycamore circuits [3] has $53$ qubits and $m$ cycles, $m = 12, 14, 16, 18, 20$. As shown in Fig. 3, each cycle has one layer of random single-qubit gates and one layer of two-qubit gates. In different cycles, the two-qubit gates are applied to different pairs of quantum qubits.

The results are summarized in Table 3. Specifically, the RL-Ising method outperforms the RL-TNCO method with a speedup of $2.84\times$. With 20 cycles, our RL-Ising method has a speedup of $4.64\times$ over the CTG-Kahypar method, $5.40\times$ over the ACQDP method, and

Table 3: Number of multiplications for Google's Sycamore circuits.

| Cycles | $m = 12$ | $m = 14$ | $m = 16$ | $m = 18$ | $m = 20$ |
|---|---|---|---|---|---|
| Scale | $\times 10^{10}$ | $\times 10^{12}$ | $\times 10^{13}$ | $\times 10^{16}$ | $\times 10^{18}$ |
| OE-Greedy [13] | $6.23 \times 10^7$ | $4.77 \times 10^7$ | $7.74 \times 10^{12}$ | $6.21 \times 10^{10}$ | $9.59 \times 10^8$ |
| CTG-Greedy [18] | $1.16 \times 10^7$ | $1.91 \times 10^7$ | $1.42 \times 10^{10}$ | $3.71 \times 10^7$ | $4.19 \times 10^7$ |
| CTG-Kahypar [18] | $2.55 \times 10^3$ | $1.41 \times 10^2$ | $1.03 \times 10^4$ | 48.0 | 6.69 |
| ACQDP [21] | $1.09 \times 10^3$ | 71 | $1.15 \times 10^4$ | 25.8 | 6.65 |
| RL-TNCO [34] | 5.44 | 7.39 | - | - | 3.49 |
| RL-Ising | **1.31** | **1.07** | **9.27** | **12.98** | **1.23** |

$2.42\times$ over the RL-TNCO method. The estimated running time of ACQDP is 21 days [21, 22]; therefore, the estimated running time of our RL-Ising method is $21/5.40 \approx 3.9$ days.

We provide a comparison of different methods in Fig. 7. With an increasing number of cycles, the number of multiplications in the log scale increases linearly, which corresponds to exponential trends (as illustrated in Fig. 1). Both RL methods, the RL-TNCO method and our RL-Ising method, are much lower than existing heuristic methods. Furthermore, the RL-Ising method is lower than that of the RL-TNCO method, with an improvement of $0.384$ orders.

**Take-home message I**: *The "quantum supremacy" claim still lacks an unequivocal first demonstration, since Google's announcement [3] was under serious questions due to the debatable estimate of $10,000$ years' running time for the classical simulation task on the Summit supercomputer*.

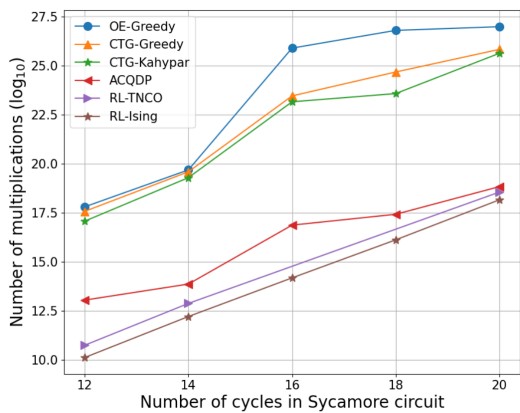

Figure 7: Number of multiplications (in log-scale) by different methods.

Table 4: Number of multiplications on large-scale quantum circuits.

| Qbits | 100 | 200 | 300 | 400 | 500 |
|---|---|---|---|---|---|
| Scale | $\times 10^{21}$ | $\times 10^{23}$ | $\times 10^{25}$ | $\times 10^{28}$ | $\times 10^{33}$ |
| OE-Greedy [13] | $8.31 \times 10^9$ | $2.26 \times 10^9$ | $6.90 \times 10^{11}$ | $7.55 \times 10^{11}$ | $4.34 \times 10^9$ |
| CTG-Greedy [18] | $3.34 \times 10^8$ | $4.24 \times 10^8$ | $5.37 \times 10^9$ | $4.49 \times 10^9$ | $1.38 \times 10^9$ |
| CTG-Kahypar [18] | $3.96 \times 10^4$ | $2.31 \times 10^4$ | $3.51 \times 10^3$ | $6.25 \times 10^2$ | $4.18 \times 10^2$ |
| **RL-Ising** | **3.85** | **5.12** | **2.27** | **3.98** | **1.84** |

## 5.4 Scale Up to Large-Scale Quantum Circuits

Scalability is very important for evaluating future hardware developments. We scale up to large-scale quantum circuits with $100, 200, 300, 400$ and $500$ qubits, respectively. There are 20 cycles for each generated quantum circuit, taking a similar structure as in Fig. 3. As given in Table 4, our RL-Ising method surpasses all the baseline methods with 227 speedups. With increasing the scale of quantum circuits, the performance gap between our method and the runner-up method CTG-Kahypar becomes large. It further shows the strong scalability of our RL-Ising method.

Therefore, we believe that reinforcement learning methods have great potential for finding the best performance curve.

**Take-home message II**: *To validate "empirical quantum supremacy" for future quantum hardware developments, the machine learning community is expected to play a critical role in maintaining publicly trustable benchmark performances with open-source training datasets.*.

## 6 Conclusion and Future work

In this paper, we have demonstrated the potential of a reinforcement learning approach for the classical simulation of quantum circuits. We reported an estimated simulation time of less than $5$ days, which is a remarkable speedup of $4.62\times$ over the state-of-the-art heuristic methods. We conduct extensive experiments to evaluate the classical simulation performance. Moreover, we have developed parallel gym environments and benchmarks, which are openly accessible as open-source resources.

However, the provided environments may not cover all types of classical simulation tasks. The best performance is an open question, which asks for continuing efforts. This project may initiate long-term collaborations from the AI/ML and quantum physics communities to maintain reference curves for validating the "empirical quantum supremacy" and drive continuing hardware advancements.

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

# Appendix A  Classical simulation task

It should be needed to sample one million samples, achieving the required XEB.

Fig. 3 shows a corresponding circuit example. It has $n = 2$ qubits for the initial state $|\psi_0\rangle$ and $m = 2$ cycles. The sampling process using the quantum circuit is computed as follows,

1. In the first cycle, random selected single-qubit quantum gates are first applied to all the four qubits of $|\psi_0\rangle$; Then, the double-qubit quantum gates are applied to $\boldsymbol{R}_1^0 |\psi_0\rangle_{0,1}$ and $\boldsymbol{R}_1^2 |\psi_0\rangle_{2,3}$, respectively, and obtain $|\psi_1\rangle$;

2. In the second cycle, random selected single-qubit quantum gates are first applied to all the four qubits of $|\psi_1\rangle$; Then, the double-qubit quantum gates are applied to $\boldsymbol{R}_2^1 |\psi_1\rangle_{1,2}$, and obtain $|\psi_2\rangle$;

3. Next, random selected single-qubit quantum gates are first applied to all the four qubits of $|\psi_2\rangle$, and obtain $\boldsymbol{R}_2^j |\psi_1\rangle_j$, $j = 0, 1, 2, 3$;

4. Last, we have a measurement $\alpha_{i_1 i_2 i_3 i_4} = \langle i_1 i_2 i_3 i_4 | \psi_3\rangle$. The sampled output is a bit-string $\alpha_{i_1 i_2 i_3 i_4}$.

---

**Algorithm 1** Random circuit sampling

---

1: **Input**: initial state $|\psi_0\rangle$, number of cycles $m$, single-qubit quantum gate $\{\sqrt{\boldsymbol{X}}, \sqrt{\boldsymbol{Y}}, \sqrt{\boldsymbol{W}}\}$, double-qubit quantum gate $\boldsymbol{U}_i$,
2: **for** $i = 1, ..., m$ **do**
3:     **for** $j = 0, ..., n - 1$
4:         $\boldsymbol{R}_i^j \leftarrow$ randomly select from $\{\sqrt{\boldsymbol{X}}, \sqrt{\boldsymbol{Y}}, \sqrt{\boldsymbol{W}}\}$,
5:         $|\psi_i\rangle_j = \boldsymbol{R}_i^j |\psi_{i-1}\rangle_j$,
6:     **end for**
7:     **for** $j = (i+1)\%2, ..., n/2 - 1$
8:         $p \leftarrow$ compute the manipulate quantum bit index $2j + (i+1)\%2$
9:         $|\psi_i\rangle_{p,p+1} = \boldsymbol{U}_i^p |\psi_{i-1}\rangle_{p,p+1}$,
10:    **end for**
11: **end for**
12: **for** $j = 0, ..., n - 1$
13:     $\boldsymbol{R}_{m+1}^j \leftarrow$ randomly select from $\{\sqrt{\boldsymbol{X}}, \sqrt{\boldsymbol{Y}}, \sqrt{\boldsymbol{W}}\}$,
14:     $|\psi_{m+1}\rangle_j = \boldsymbol{R}_{m+1}^j |\psi_m\rangle_j$,
15: **end for**
16: Obtain a measurement by $\alpha_{i_1 i_2 ... i_n} = \langle i_1 i_2 ... i_n | \psi_{m+1}\rangle$,
17: **Output**: a bit-string $\alpha_{i_1 i_2 ... i_n}$.

---

**Quantum circuits**: There have been many quantum circuits proposed as follows,

- Sycamore quantum [3]: It consists of 53 qubits and 20 cycles. For the Boson sampling problem, it only needs 200 seconds to finish this task, while it needs $10,000$ years for classical simulation.

- Jiu Zhang [60]: It consists of 74 qubits. For the Gaussian Boson Sampling problem, it can use 200s to finish up to a million times compared with classical simulations.

- Zuchongzhi [61]: It has 60 qubits with the number of 24. The achieved sampling task is about 6 orders of magnitude more difficult than that of Sycamore in the classic simulation.

# Appendix B  Data Generators and Reinforcement Learning Environments.

## B.1  Tensor Network Representations

We use the unidirectional graph to represent the tensor network. Specifically, for a given tensor network with $n$ tensors, we use a symmetric $n \times n$ matrix $\boldsymbol{M}$ to represent the dimensional relationships between tensors, which is named the dimension matrix. Specifically, if the tensors with index $i$ and $j$ are connected with a shared dimension as $d_{ij}$, then we set $\boldsymbol{M}_{ij} = \boldsymbol{M}_{ji} = d_{ij}$, otherwise, it is

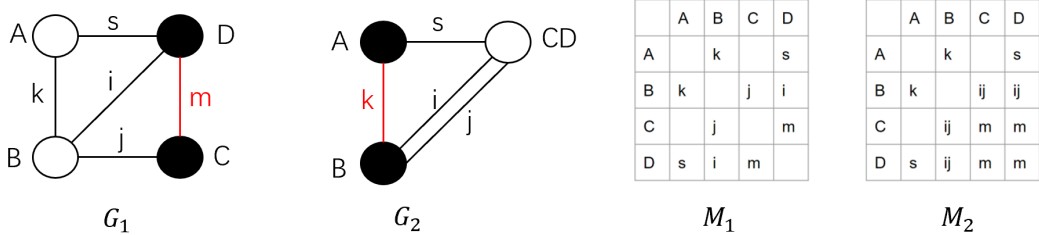

Figure 8: Example of the tensor network contraction environment.

set as $0$. We also denote a connection matrix $C$, where $C_{ij} = C_{ji} = 1$ indicates the connection between the $i$-th tensor and $j$-th tensor.

## B.2 Tensor Network Contraction

We employ tensor network contraction as the underlying computational framework in our quantum circuit simulation. Given a specific tensor network structure and a specified contraction order, the environment is designed to provide the number of multiplications required during the tensor contraction process.

Specifically, as illustrated in Figure 8, we consider a tensor network comprising $4$ tensors connected in a grid structure. When performing a tensor contraction between tensors $C$ and $D$, it is necessary to compute the number of multiplications involved. From the unidirectional graph representation, tensor $C$ has dimensions $M \times J$, tensor D has dimensions $S \times I \times M$, and a shared edge with dimension $M$ exists.

To contract tensors $C$ and $D$ along the edge with dimension $M$, the number of multiplications can be computed as $\frac{(MJ) \times (SIM)}{M}$. This expression takes into account the multiplication of the dimensions $MJ$ from tensor C and $SIM$ from tensor D, divided by the shared dimension $M$.

Upon completing the contraction, it is essential to update the unidirectional graph representation. The contracted tensor formed by the contraction of C and D retains an independent edge with dimension $M$ ($M_{33} = M_{34} = M_{43} = M_{44} = M$). The connected edges between the contracted tensor ($C$ or $D$) and other tensors are determined by multiplying the original edge dimensions between them ($M_{23} = M_{24} = M_{32} = M_{42} = IJ$). This update to the unidirectional graph ensures accuracy and reflects the changes resulting from the contraction step.

By incorporating the update of the unidirectional graph and computing the number of multiplications, we complete the current simulation step.

## B.3 Verification

**Unified representation**: We use a tuple consisting of the contracted tensor indices to represent the tensor contraction order. The resulting tensor is labeled as a new index. As shown for example in Fig. 5, we first write down $(3, 4)$ for the first contraction operation. We label the generated tensor $34$ as $5$. Next, we use $(1, 2)$ to label the second tensor contraction operation and $6$ to label the resulting tensor $12$. Last, we contract the tensor $12$ and $34$, where we use $(5, 6)$ to represent the contraction record. Thus, we have the contraction order $\{(3, 4), (1, 2), (12, 34)\}$

**Calculating the number of multiplications**: Given the tensor contraction order, we will compute the number of involved multiplications. In Fig. 5, using different optimizers, we can get a specific contraction order, like $\{(3, 4), (1, 2), (12, 34)\}$. We first contract the 3-th and 4-th tensor, of which the number of multiplication is $IJMS$. Then, for the contraction order $(1, 2)$, we have the number of multiplication as $SKIJ$. Last, for the contraction order $(12, 34)$, we have the number of multiplication as $SIJ$.

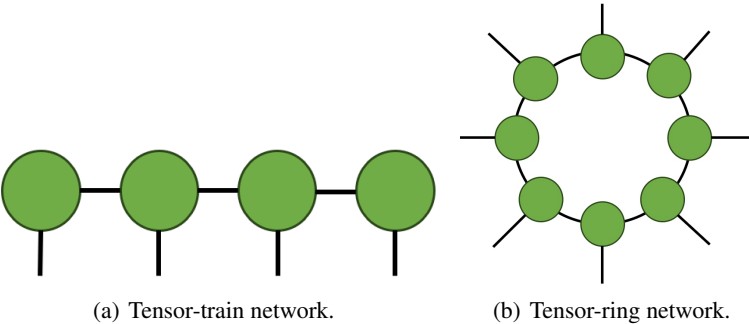

(a) Tensor-train network.   (b) Tensor-ring network.

Figure 9: Illustration of the tensor train and tensor ring network.

## B.4    Datasets for Different Tensor Networks

In this section, we will first introduce the data structure used to describe the relationships between different tensor nodes. Then, we will demonstrate the generation process of the dataset for different tensor networks.

**Data Structure**: The tensor network can be represented as an undirected graph $G = \{V, E, w\}$ and is amenable to be stored using the adjacent table, which can be defined as follows (C-format),

```
class TensorNetwork {
    int V; // number of tensor nodes
    int E; // number of edges
    Queue[] adj; // adjacent nodes for each node

    TensorNetwork(int V) { // init the graph
        this.V = V; this.E = 0;
        this.adj = new Queue[V];
        for (i = 0; i < V; i++) {
            this.adj[i] = new Queue[];
        }
    }
    void AddEdge(int v, int w) { // add the edges
        if (w > v) v, w = w, v;
        this.adj[v].enqueue(w);
        this.E++;
    }
    void Build(...); //differs in different tensor networks
}
```

Specifically, we use $V$ and $E$ to save the number of tensor nodes and edges, respectively. The variable $adj$ is a queue with flexible length, where $adj[j]$ is also a queue to save the adjacent tensor nodes. Given the total number of tensor nodes $V$, $TensorNetwork(V)$ is used to initialize the undirected graph, where each tensor node is assigned an empty queue. The $addEdge(\cdot, \cdot)$ is invoked to add the connected relationship between the connected tensor nodes $v$ and $w$. We make a simplified assumption that only the larger index $w$ can be added to the adjacent queue of $v$. Different tensor network differs in the implementation of the $Build(...)$ function, which invokes the $addEdge(\cdot, \cdot)$ depending on the specified tensor network structure.

**Tensor-Train Network or Matrix Product States**: The matrix product state (MPS) [41, 56] or tensor-train (TT) represents a tensor as a chain-like contraction of third-order tensors with the head and tail as matrices. The tensor diagram of MPS/TT is shown in Fig. 9(a).

MPS has been widely used in modeling quantum circuits such as [47, 16, 45]. In the $Build(...)$ function, for each tensor node $i$, we need to use the $AddEdge(i, i + 1)$ to add the $i + 1$-th node to the adjacent lists of $i$-th node, $i < |V| - 1$. We can set up different number of tensor nodes $N$ to generate the dataset for MPS tensor network.

```
    void TensorNetwork::Build(){ // Build for MPS
        for (i = 0; i < this.V-1; i++) this.AddEdge(i, i+1); }
```

The generation code for the tensor-train network is saved in `https://github.com/XiaoYangLiu-FinRL/RL4QuantumCircuits/blob/main/datasets/mps/mps_generate.py`.

**Tensor ring network**: The tensor ring (TR) represents a high-order (or high-dimensional) tensor by a sequence of 3rd-order tensors that are multiplied circularly, whose tensor diagram notation can be represented in Fig. 9(b).

The tensor-ring network has been utilized to simulate the quantum circuit in [44].

The main difference between the tensor ring and the MPS tensor network is that the first and last tensor nodes in tensor ring network are also connected. Thus, only a minor modification of MPS generation can be applied to generate the tensor ring network.

```
void TensorNetwork::Build(){ // Build for tree tensor
    for (int i = 0; i < this.V; i++)
        this.AddEdge(i, (i+1)%this.V); }
```

The generation code for the tensor-ring network is saved in `https://github.com/XiaoYangLiu-FinRL/RL4QuantumCircuits/blob/main/datasets/tr/tr_generate.py`.

**Tree Tensor Network**: Tree tensor network (TTN) [37, 52, 55] or Hierarchical Tucker (HT) is a generalization of MPS that encodes a tree entanglement structure. The diagram notation of a TTN can be represented in Fig. 10.

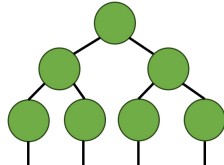

Figure 10: Tree tensor network.

Tree tensor network has been used to model the quantum [24, 50], and quantum chemistry [36, 37].

Tree tensor network is a fully binary tree structure, of which the number of tensor nodes depends on the height. we reload the initialization function and write the build function as follows,

```
TensorNetwork::TensorNetwork(int H) { // init the graph
    this.H = H; this.V = pow(2,H) - 1; this.E = 0;
    this.adj =  new Queue[V];
    for (i = 0; i < V; i++) {
        this.adj[i] = new Queue[];
    }
}
void TensorNetwork::Build(){ // Build for tensor ring
    this.AddEdge(0, 1); this.AddEdge(0, 2);
    for (h = 1; h < this.H; h++){
        for (v = pow(2, h-2); v < pow(2, h-1) + 1; v++){
            this.AddEdge(v, 2v); this.AddEdge(v, 2v+1);
        }
    }
}
```

The generation code for the tree tensor network is saved in `https://github.com/XiaoYangLiu-FinRL/RL4QuantumCircuits/blob/main/datasets/tree/tree_generate.py`.

**PEPS Network**: The PEPS (projected entangled pair state) tensor network [49, 58] generalizes MPS from a one-dimensional network to a network on an arbitrary graph, whose tensor diagram notation can be represented in Fig. 11.

Some work applies the PEPS network to quantum circuits [43] and quantum systems [40, 31].

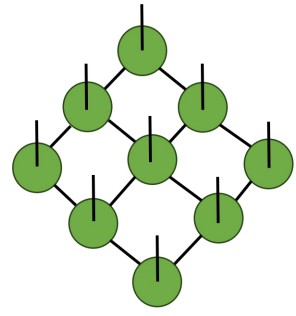

Figure 11: PEPS tensor network.

The generation code for the PEPS network is saved in `https://github.com/XiaoYangLiu-FinRL/RL4QuantumCircuits/blob/main/datasets/peps/peps_generate.py`.

**MERA Network**: The MERA (Multiscale Entanglement Renormalization Ansatz) [14] tensor network colleagues as a refinement of the MPS and PEPS. It has a hierarchical structure, with layers of tensors representing increasingly coarse-grained degrees of freedom, which can be represented in Fig. 12.

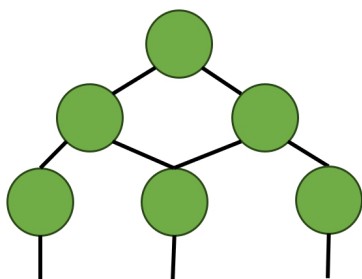

Figure 12: MERA tensor network.

Some work applies the MERA network to quantum circuits [1, 20, 1, 30].

The generation code for the MERA tensor network is saved in `https://github.com/XiaoYangLiu-FinRL/RL4QuantumCircuits/blob/main/datasets/mera/mera_generate.py`.

**Google's Sycamore Circuits**: Some work has put efforts into transforming the Sycamore circuits into tensor representations. The provided data is usually organized into the adjacent graph structure, while the indices of the $i$-th row are the indices of connected tensors to $i$-the tensor. The tensor network representation of the Sycamore circuits corresponds to a complicated net structure. To map the Sycamore circuits into the tensor network environment, we need to iteratively read each line of the given Sycamore file and fill it up with corresponding dimension information.

The generation code for the Sycamore circuit is saved in `https://github.com/XiaoYangLiu-FinRL/RL4QuantumCircuits/tree/main/datasets/sycamore`.

## B.5 Environment

We provide the gym-environment for the tensor network contraction problem in our classical simulation of quantum circuits. The code is provided in `https://github.com/XiaoYangLiu-FinRL/RL4QuantumCircuits/blob/main/rl/mps/env.py` for tensor-train tensor network and `https://github.com/XiaoYangLiu-FinRL/RL4QuantumCircuits/blob/main/rl/sycamore/env.py` for Sycamore circuits.

Specifically, the gym-environment for the tensor network contraction problem in our classical simulation of quantum circuits is designed as follows,

- Init/Restart: initialize the states of environments, including the number of nodes, the number of edges, *etc.*
- Step: Given the tensor network contraction order, compute the number of involved multiplications as the reward. The rewards are represented in log scale.

The implementations can be found in *init()*, and *get_log10_multiple_times()* functions of our *env.py*.

## B.6 High-Performance Reinforcement Learning for TNCO

**Mapping onto the K-spin Ising Model**: We use the K-spin Ising model to formulate the TNCO problem as follows,

$$
H(x) = \sum_{i=1}^{N-1} \left[ \left( 2 - \sum_{u=1}^{N-i} x_{u,i} \right)^2 + \sum_{u=1}^{N} \sum_{v=1}^{N} J_{u_i,v_i} x_{u,i} x_{v,i} \right],
$$

$$
H(x^1, ..., x^K) = \sum_{k=1}^{K} H(x^k) + \sum_{k=1}^{K} \sum_{i_1 \in V^1} ... \sum_{i_1 \in V^k} J_{i_1...i_k} x_{i_1}^1 ... x_{i_k}^k,
$$
(12)

where we denote the $N(N-1)$ spin as $x_{u,j}$, $u$ as the tensor and $j$ denotes its order in the TNCO path.

We use the variational annealing methods to solve the K-spin Ising model problem of TNCO. Specifically, we minimize the KL divergence between the transition distribution of $x$ to $x'$ with the target Boltzmann distribution as follows,

$$
\begin{aligned}
\mathcal{D}_{KL}(q_\theta||p) &= \sum_x q_\theta(x \to x') \ln \left( \frac{q_\theta(x \to x')}{p(x \to x')} \right) \\
&= \sum_x q_\theta(x \to x') \ln q_\theta(x \to x') \\
&+ \frac{q_\theta(x \to x')(H(x) - H(x'))}{T} + \ln Z_x,
\end{aligned}
$$
(13)

where $T$ is the temperature, $q_\theta$ is the distribution of TNCO path $x \to x'$ parameterized by $\theta$, $p(x'|x)$ is the transition distribution of Boltzmann distribution $p(x'|x) = \frac{\exp(-\frac{H(x')}{T})}{Z}$, $Z_x = \sum_{x'}^{H(x')<H(x)} e^{-\frac{H(x')}{T}}$. During the learning process, we gradually anneal the temperature to optimize (13).

**Parallel data sampling**: We use RNN parameterized by $\theta$ as the policy network to model the transition probability $q_\theta(x \to x')$. We input the K-spin representation of tensor contraction order, $(x_1, ..x_K)$, sequentially from left to right, and compute the transition probability as follows,

1. Randomly initialize the hidden variable $h_1$;
2. Input $x_1$ and $h_1$ to the RNN, and output the next hidden variable $h_2$ and a transition probability $q_\theta(x_1)$;
3. Input $x_2$ and $h_2$ to the RNN, and output the next hidden variable $h_3$ and a transition probability $q_\theta(x_1 \to x_2)$;
4. ......
5. Input $x_K$ and $h_K$ to the RNN, and output the next hidden variable $h_{K+1}$ and a transition probability $q_\theta(x_{K-1} \to x_K)$.

Then, by sampling the tensor contraction ordering trajectories, $(x_1, x_2, ..., x_K)$, we can optimize (13) to train the RNN. The sampling process can be batched onto multiprocess to achieve high-performance data sampling, thus alleviating the performance bottleneck of reinforcement learning training.

**Parallel training**: We initialize multiple optimizers in parallel to learn to optimal tensor contraction order. Specifically, at the $t$-th iteration

1. We parallelly sample $N$ tensor contraction ordering from the TNCO environment and store them in the reply buffer.

2. For the optimizer $i$ -th, we first freeze $\rho_{t,i}$ parameters, $0 < \rho_i < 1$, then sample data from replay buffer, and compute the loss function (13) with the temperature $T_{t,i}$, independently.

3. For $i$-th optimizer, we compute the gradient and use LSTM to learn to optimize the parameter.

We vary the parameter masked ratio $\rho$ and temperature $T$ by increasing the number of iterations and initialized all the parameters differently to achieve swarm intelligence integrated with the curriculum learning.

## Appendix C  Accessibility, Usage, License, and Maintenance

**Accessibility**: All the code, dataset, and tensor network contraction orderings, including the Sycamore circuits, can be found in our open source project `https://github.com/XiaoYangLiu-FinRL/RL4QuantumCircuits` without personal request.

**Dataset generation**: We generate the classical simulation of quantum circuits synthetically using `https://github.com/XiaoYangLiu-FinRL/RL4QuantumCircuits/tree/main/datasets`.

For example, we run `https://github.com/XiaoYangLiu-FinRL/RL4QuantumCircuits/blob/main/datasets/mps/mps_generate.py` to generate quantum circuits based on the tensor train, where $V$ in code controls the number of nodes in generated data.

**Code organization**: We implement the baseline methods using Opt-einsum and Cotengra to search for the tensor network contraction orderings. The codes for calling these solvers are provided in `https://github.com/XiaoYangLiu-FinRL/RL4QuantumCircuits/tree/main/baseline`. We provide two methods to solve each type of quantum circuit, like tensor train-based quantum circuits.

**Usage**: To run the baseline methods, execute "python cotengra.py" or "python opt_einsum.py", where the variable n is the number of nodes. To generate the dataset, please execute "python generate.py", where the variable $V$ in the central part is the number of tensor nodes.

**License**: MIT License.

**Maintenance**: On GitHub, we keep updating our codes, merging pull requests, and fixing bugs and issues. We welcome contributions from community members and researchers.

