# OpenReview forum: "Classical Simulation of Quantum Circuits: Parallel Environments and Benchmark"
_NeurIPS.cc/2023/Track/Datasets_and_Benchmarks — NeurIPS 2023 Datasets and Benchmarks Poster_

### Official Review · Reviewer_m3Yr · 2023-07-20
**Review comments**

**Rating:** 7
**Confidence:** 3
**Correctness:** Yes
**Clarity:** Yes

**Strengths:**

The concept of quantum supremacy is indeed a significant and contemporary topic in the field of quantum computing. If true quantum advantage is achieved, it could herald a new era of computation, potentially revolutionizing fields such as cryptography, optimization, and material science. The proposed benchmark could provide a more transparent and trustable way to validate empirical quantum supremacy.

This work proposes a reinforcement learning method that achieves a remarkable speedup over SOTA heuristic methods, which demonstrates potential of RL for the classical simulation of quantum circuits. In addition, five baselines are provided in benchmark, which offers a solid, meaningful reference for subsequent research in this field. Furthermore, a pool of implementation tricks is also provided to overcome challenges for developing RL algorithms.

Overall, I believe the contribution of this work is significant for its research topic, solid experiments, and well-organized GitHub repo.


**Additional Feedback:**

N/A

**Documentation:**

Yes

**Opportunities For Improvement:**

I’m a little bit confused about the “standard criteria to evaluate the performance of classical simulation of quantum circuits”. Do the standard criteria comprise experiments shown in Sec 5.2, 5.3 and 5.4? What’s the metric for scalability? It would be easier for readers who’s not an expert of quantum physics to understand if authors can summarize the standard evaluation you recommend in Sec 5.

As mentioned in Sec 6, not all types of classical simulation tasks are covered in the provided environments. In the future, more tasks can be added and long-term collaborations from the AI/ML and quantum physics communities are encouraged.


**Relation To Prior Work:**

Yes

**Summary And Contributions:**

This paper formulates a tensor network contraction ordering problem and proposes a dataset with a benchmark for this classical simulation of quantum circuits. In addition to several baseline methods, authors also propose a reinforcement learning approach for fast simulation.

---

> ### Author Response · Authors · 2023-08-28
>
> **Q1: I’m a little bit confused about the “standard criteria to evaluate the performance of classical simulation of quantum circuits”. Do the standard criteria comprise experiments shown in Sec 5.2, 5.3 and 5.4? What’s the metric for scalability? It would be easier for readers who’s not an expert of quantum physics to understand if authors can summarize the standard evaluation you recommend in Sec 5.**
>
> __A1:__ Mathematically, the classical simulation of quantum circuits can be mapped into the tensor network contraction problem. The simulation time can be estimated by the number of multiplications involved in the tensor network contraction [1][2]. We denote __the number of multiplications__ involved in the tensor network contraction as the standard criteria for the simulation performance. Therefore, the target problem becomes an NP-hard problem, which can be viewed a variant of travelling salesman problem in the context of simulating quantum circuits on electrical computers.
>
> We vary the number of qubits in classical simulation of quantum circuits, i.e., __the number of nodes in the tensor network__, to evaluate the scalability. In our work, we evaluate the performance of our proposed methods on various tensor networks with a range of nodes. For the tensor-train network, we vary the number of tensors from 400 to 2000. For the synthetic random tensor networks, we vary the number of nodes from 20 to 100, and the degrees of individual nodes  from the 2 to 6. For large-scale quantum circuits, we increase the number of qubits from 100 to 500 to evaluate the scalability of our proposed method.
>
> [1] Huang, Cupjin, et al. "Classical simulation of quantum supremacy circuits." arXiv preprint arXiv:2005.06787 (2020).
> [2] Meirom, Eli, et al. "Optimizing tensor network contraction using reinforcement learning." International Conference on Machine Learning. PMLR, 2022.
>
>
> **Q2: As mentioned in Sec 6, not all types of classical simulation tasks are covered in the provided environments. In the future, more tasks can be added and long-term collaborations from the AI/ML and quantum physics communities are encouraged.**
>
> __A2:__ Thanks for your advice! In this work, we study on the “quantum supremacy” circuit, which lacks an __unequivocal first demonstration__. Google's work [1] used the random number generation problem for the statement of "Quantum supremacy" [1].   We follow Google's setting and accelerate the classical simulation of quantum circuits on electrical computers.
>
> We agree that more simulation tasks can be included to provide more oppotunities for the collaborations from the AI/ML and quantum physics communities.  We will include more types of classical simulation tasks in our environments, including the the traveling-salesman problem [2], graph maxcut [3], etc [4,5,6].
>
> [1] Arute, Frank, et al. "Quantum supremacy using a programmable superconducting processor." Nature 574.7779 (2019): 505-510.
> [2] Martoňák, Roman, Giuseppe E. Santoro, and Erio Tosatti. "Quantum annealing of the traveling salesman problem." Physical Review E 70.5 (2004): 057701.
> [3] Wang, Zhihui, et al. "Quantum approximate optimization algorithm for MaxCut: A fermionic view." Physical Review A 97.2 (2018): 022304.
> [4] Xu, Xiaosi, et al. "A Herculean task: Classical simulation of quantum computers." arXiv preprint arXiv:2302.08880 (2023).
> [5] Napp, John C., et al. "Efficient classical simulation of random shallow 2D quantum circuits." Physical Review X 12.2 (2022): 021021.
> [6] Ravi, Gokul Subramanian, et al. "CAFQA: A classical simulation bootstrap for variational quantum algorithms." Proceedings of the 28th ACM International Conference on Architectural Support for Programming Languages and Operating Systems, Volume 1. 2022.

---

### Official Review · Reviewer_YFU3 · 2023-07-21
**Good ML Approaches but Limited Insights on Data**

**Rating:** 6
**Confidence:** 3
**Clarity:** Overall, the paper is well written.

**Strengths:**

The paper explains the basic concepts of quantum circuits in a clear manner for non-quantum experts.
The training algorithm and reinforcement learning approach was clearly articulated, and has lead to strong results compared to other methods on both the proposed synthetic dataset and on Google's Sycamore circuits.

**Additional Feedback:**

Some notes/questions on the paper:
- Figure 1: please label the y-axis. Also, what does "10,000 years" and "21 days" refer to? The required time by classical computers to solve the problem?
- Line 136: Should there be an equal sign between psi_i_j and R_i*psi_(i-1)_j?
- Line 243: Please cite other papers that use swarm intelligience for optimization
- Lines 254-256: Please clarify why large number of training iterations leads to insufficient optimization
- Lines 258-259: Freezing parameters during training and unfreezing them gradually is interesting. Personally I haven't heard of that curriculum approach. If it is done in a previous paper, please cite, otherwise it could be a contribution of the paper
- Line 259: "freeze 90% parameters" should be "freeze 90% of the parameters"?
- Line 277: typo: "slicin g.," should be "slicing"

**Correctness:**

The dataset link points to a script that generates the dataset: https://github.com/XiaoYangLiu-FinRL/RL4QuantumCircuits/tree/main/datasets/mps
but the script looks quite simple with no arguments, seed values, etc. to reproduce the same dataset as the paper evaluated.

Also, I suggest to upload the actual dataset (in addition to the script and arguments that generated it) so that other researchers can reproduce the numbers or evaluate it.

**Documentation:**

Insufficient detail on data generation, licensing, or maintenance was provided.
A link to a script that generates the dataset was provided, but the arguments provided to the script or the random seed value wasn't provided. Hence, no sufficient detail was provided to support reproducibility.
The link to the actual dataset isn't provided.

**Limitations:**

Very limited information were provided on the dataset, while the paper was submitted to a Datasets Track.

**Opportunities For Improvement:**

More details and analysis are required about the dataset, as this paper is submitted to a Datasets Track. Some ideas that could be analyzed:
- what are the advantages of this dataset compared to Google's Sycamore Circuits?
- does the dataset provide insights or indicate differences between simulation approaches that are not indicated by previous datasets?
- Perhaps run ablation studies to analyze the relationship between average runtime and criteria like number of nodes, edges, etc.

**Relation To Prior Work:**

The paper explained the differences between its model or training algorithm compared to previous classical simulation techniques, but didn't explain the difference between its dataset and previous datasets.

**Summary And Contributions:**

Tensor networks or quantum circuits take a long time to simulate, and are quickly executed by quantum computers. However, classical simulation techniques of such circuits are becoming faster and faster, and hence claims by companies like Google that they have created a quantum computer that executes a circuit that requires 10,000 years by a classical computer to simulate is questionable.
Hence, there is a need for a dataset of tensor networks to evaluate the performance of classical simulation techniques and create a common benchmark for researchers to compare against.
This paper presents a synthetic dataset for tensor networks to evaluate the runtime of classical simulation techniques and compare them to claims by Google about their quantum circuit superiority.
The paper also presents a reinforcement learning training recipe, RL-Ising that beats other classical simulation approaches in terms of runtime.

---

> ### Author Response · Authors · 2023-08-25
>
> **Q1: What are the advantages of this dataset compared to Google's Sycamore Circuits?**
>
> __A1__: Compared to Google's Sycamore circuits [1], there are three advantages in our dataset,
> 1. **Provided a dozens of quantum circuits in gym-style environments.** Our dataset is larger, which not only includes Google's Sycamore circuits, but also includes quantum circuits represented by various tensor networks, including the  matrix product states, tensor ring network, tree tensor network, PEPS network,  MERA network, and random tensor networks.
> 2. **Diverse benchmarking methods.** We provide more benchmark methods, including the OE-greedy, CTG-Greedy, CTG-Kahypar, and RL-TNCO,   and design a unified representation of the tensor network contraction order found by different methods. This design benefits researchers to compare different methods and verify the contraction order.
> 3. **High-performance gym-style RL environments** We provide gym-style environments for RL algorithms, which supports __massive parallel classical simulaiton__ of quantum circuits. It is useful for researchers to develop advanced algorithms efficiently.
>
> [1] Arute, Frank, et al. "Quantum supremacy using a programmable superconducting processor." Nature 574.7779 (2019): 505-510.
>
> **Q2: does the dataset provide insights or indicate differences between simulation approaches that are not indicated by previous datasets?**
>
> __A2__:  In this project, we propose to leverage reinforcement learning methods to establish benchmark performance of the classical simulation of quantum supremacy circuits.  For the random number generation task, Google’s “quantum supremacy [1]” announcement relied on an wrongly estimated simulation time of 10 000 years on the Summit supercomputer. In our updated version, we further reduce it to less than 4 days. That empirical claim of quantum supremacy by Google team [1] should have been rejected by academia, and such a “quantum supremacy” claim shall be viewed as Google’s hasty advertisement. Our conclusion is that the search of “quantum supremacy” still lacks an unequivocal first demonstration.
>
> [1] Arute, Frank, et al. "Quantum supremacy using a programmable superconducting processor." Nature 574.7779 (2019): 505-510.

---

> > ### Author Response · Authors · 2023-08-25
> >
> > **Q3: Perhaps run ablation studies to analyze the relationship between average runtime and criteria like number of nodes, edges, etc.**
> >
> > __A3__: Thanks for your suggestion! We provided the experiments on the scalability in Section 5.2, where we vary the number of qubits (nodes) of the synthetic tensor-train networks in Table 1.
> >
> > We conduct the ablation study on K  in our K-spin Ising model.  Specifically, we vary the value of K from 1 to 5 in the K-spin Ising model for the classical simulation of Sycamore circuit, as shown in Table 1.
> > > Table 1: Varying K from 1 to 5, the number of multiplications for the classical simulation of Sycamore circuits.
> > > | K           | m=12               |  m=14         | m=16     |m=18     |m=20     |
> > > | :---        |    :----:          |          ---: | ---:     | ---:    |---:     |
> > > | 1           |$1.38\times 10^{10}$ |  $1.15\times 10^{12}$ |  $9.35\times 10^{13}$    |  $13.19\times 10^{16}$  |    $1.45\times 10^{18}$    |
> > > | 2           |$1.36\times 10^{10}$ |  $1.09\times 10^{12}$ |  $9.28\times 10^{13}$    |  $13.05\times 10^{16}$  |    $1.44\times 10^{18}$    |
> > > | 3           |$1.31\times 10^{10}$ |  $1.07\times 10^{12}$ |  $9.27\times 10^{13}$    |  $12.98\times 10^{16}$  |    $1.42\times 10^{18}$    |
> > > | 4           |$1.31\times 10^{10}$ |  $1.06\times 10^{12}$ |  $9.27\times 10^{13}$    |  $12.98\times 10^{16}$  |    $1.42\times 10^{18}$    |
> > > | 5           |$1.31\times 10^{10}$ |  $1.06\times 10^{12}$ |  $9.27\times 10^{13}$    |  $12.97\times 10^{16}$  |    $1.42\times 10^{18}$    |
> >
> >
> >
> > From the result, we can see that it has a large performance improvement with increasing K from 1 to 3, while it converges when further increase the K value. It indicates that the introduction of K steps accelerate the convergence by considering more steps in the trajectory.
> >
> >
> > We also conduct ablation study on the number of edges. For the classical simulation of synthetic random tensor networks, we vary the edges from 2 to 5, the qubits from 25 to 100, and use our method to search the optimal contraction path. The results are summarized in Table 2. It can be seen that, the number of multiplication involved increases with the increasing number of edges and qubits.
> > > Table 2: Varing the degree of individual nodes from 2 to 5, the qubits from 25 to 100, the number of multiplications for   synthetic random tensor networks.
> > > | Qubits           | d=2               |  m=3         | m=4     |m=5     |
> > > | :---        |    :----:          |          ---: | ---:     | ---:    |
> > > | 25           |$1.13\times 10^{4}$ | $1.25\times 10^{4}$  | $1.35\times 10^{4}$ | $1.41\times 10^{4}$|
> > > | 50           | $2.37\times 10^{7}$|  $2.51\times 10^{7}$ |  $2.58\times 10^{7}$|$2.69\times 10^{7}$ |
> > > | 75           |$6.85\times 10^{9}$ |  $7.00\times 10^{9}$ | $7.39\times 10^{9}$ |$7.54\times 10^{9}$ |
> > > | 100           |$4.75\times 10^{12}$ |  $4.90\times 10^{12}$ | $4.98\times 10^{12}$ | $5.12\times 10^{12}$|
> >
> >
> >
> >
> > **Q4: Very limited information were provided on the dataset, while the paper was submitted to a Datasets Track.**
> >
> > __A4__: In our appendix, we provided detailed information for the data structure and data generation of the tensor network environments. Specifically,
> > + We present the unified representation of the tensor contraction order in Appendix B.1.
> > + We present the tensor network contraction  of tensor network environment in Appendix B.2.
> > + We present the verification of the tensor network contraction order of the tensor network environment in Appendix B.3.
> > + We presnet the implementation of tensor network contraction environment and various tensor networks in Appendix B.4, including matrix product states, tensor ring network, tree tensor network, PEPS network, MERA network, and Google's Sycamore circuits.
> > + We present the design of tensor network contraction environment in Appendix B.5.
> > + The accessibility, usage, license, and maintenance are provided in Appendix C.
> >
> > In our github repo, we corresponding provide information of our dataset in the README.md file as follows,
> > + We provide the  foreword,  goal, roadmap, organization of codes, and experimental results, related blogs, and reference in https://github.com/XiaoYangLiu-FinRL/RL4QuantumCircuits/blob/main/README.md.
> > + We provide the benchmark methods for each tensor network contraction environments in https://github.com/XiaoYangLiu-FinRL/RL4QuantumCircuits/tree/main/baseline.
> > + We provide the data generation code for each tensor network contraction environment in https://github.com/XiaoYangLiu-FinRL/RL4QuantumCircuits/tree/main/datasets_generator, and save some demo examples in https://github.com/XiaoYangLiu-FinRL/RL4QuantumCircuits/tree/main/datasets_generator.
> > + We provide the optimal tensor network order of Sycamore circuits in https://github.com/XiaoYangLiu-FinRL/RL4QuantumCircuits/tree/main/verification/orders/sycamore.

---

> > > ### Author Response · Authors · 2023-08-25
> > >
> > > **Typo issues:  Some notes/questions on the paper:
> > >
> > > Figure 1: please label the y-axis. Also, what does "10,000 years" and "21 days" refer to? The required time by classical computers to solve the problem?
> > > Line 136: Should there be an equal sign $|\psi_{i}\rangle_{j}$ and $R_i |\psi_{i-1}\rangle_{j}$?
> > > Line 243: Please cite other papers that use swarm intelligience for optimization
> > > Lines 254-256: Please clarify why large number of training iterations leads to insufficient optimization
> > > Lines 258-259: Freezing parameters during training and unfreezing them gradually is interesting. Personally I haven't heard of that curriculum approach. If it is done in a previous paper, please cite, otherwise it could be a contribution of the paper
> > > Line 259: "freeze 90% parameters" should be "freeze 90% of the parameters"?
> > > Line 277: typo: "slicin g.," should be "slicing"**
> > >
> > > __A__: Thanks for your suggestion and advice. We make the following revisions:
> > > 1.  __Figure 1__: We have added a label to the y-axis indicating simulation time. Additionally, "10,000 years" and "21 days" now explicitly denote the time required by classical computers to solve the random number generation problem.
> > > 2. __Line 136__: An equal sign has been added between $|\psi_{i}\rangle_{j}$ and $R_i |\psi_{i-1}\rangle_{j}$.
> > > 3. __Line 243__: In response to the reviewer's suggestion, we have included citations to related papers utilizing swarm intelligence for optimization:
> > > > [1] Blum, Christian, and Xiaodong Li. "Swarm intelligence in optimization." Swarm intelligence: introduction and applications. Springer Berlin Heidelberg, 2008. Pages 43-85.
> > > > [2] Nayyar, Anand, Dac-Nhuong Le, and Nhu Gia Nguyen, eds. "Advances in swarm intelligence for optimizing problems in computer science." CRC Press, 2018.
> > > 4. __Lines 254-256__: We have clarified the reason behind the insufficient optimization resulting from a large number of training iterations. Specifically, we explain that a high number of steps in the sampled trajectories of the K-spin Ising model poses challenges in optimizing the original problem.
> > > 5. __Lines 258-259__: There several works using the curricultm learning for optimizaiton, including the freezimng parameters approach. We add the citation in our revison.
> > > > [1] Goutam, Kelam, et al. "Layerout: Freezing layers in deep neural networks." SN Computer Science 1.5 (2020): 295.
> > > > [2] Zhu, Qingqing, et al. "Combining curriculum learning and knowledge distillation for dialogue generation." Findings of the Association for Computational Linguistics: EMNLP 2021. 2021.
> > > 6. __Line 259__: The suggested rephrase "freeze 90% of the parameters" has been implemented to enhance clarity.
> > > 7. __Line 277__: The typo "slicin g" has been corrected to "slicing."

---

> > > ### Comment · Reviewer_YFU3 · 2023-08-31
> > >
> > > Thanks for your detailed answers to my questions.
> > >
> > > For the 2 Tables, can you clarify what symbols d, m, and K.denote?
> > >
> > > I tried looking into the simulation data, but found a text file like this:
> > > https://github.com/XiaoYangLiu-FinRL/RL4QuantumCircuits/blob/main/simulation_data/peps/V_10.txt
> > > Is the content of this file in binary or text format?

---

> ### Comment · Reviewer_YFU3 · 2023-08-31
>
> I would like to thank the authors for their detailed answers.
> I have read the other reviewers' feedback and read the authors rebuttals as well.
>
> The authors have addressed most of my concerns. I still believe the contribution is more in the RL solution rather than the dataset or benchmark.
>
> I would advise the authors to ensure that there are clear instructions in their repo to download the dataset so that others can evaluate and train their approaches on it in the future.
>
> I would advise that the repo's README contain code snippets for:
> - Gym-like APIs
> - iterating over the dataset
> - inference on a trained model
> - training a new midel
>
> I have increased my score from 4 to 6.

---

### Official Review · Reviewer_m3Yv · 2023-07-24
**Benchmark for quantum simulations using Hamiltonian-based RL and K-spin Ising model.**

**Rating:** 7
**Confidence:** 4
**Correctness:** Yes, the paper's results are consiste…

**Strengths:**

(+) The idea of using a Hamiltonian derived from the K-spin Ising model for solving the TNCO problem is interesting, and shows experimental promise based on the results of this paper.
(+) Experimentally, the code associated with the paper can simulate Google's Sycamore circuits in 5 days, while the earlier state-of-the-art takes 10 days. This is an impressive experimental performance.
(+) Five different baseline implementations are provided together with the proposed approach.
(+) The implementation details, such as L2O strategy, masking network parameters, and separate saving of high-quality tensor contractions orders in the replay buffers, are clearly discussed in the paper.

**Additional Feedback:**

Table 4 and Eqn. 8 should be more clearly discussed.

**Clarity:**

The paper is clearly written. There are very minor presentation issues but that do not conflict with the overall message of the paper.

**Documentation:**

Not Applicable

**Ethics:**

No concerns.

**Limitations:**

The paper has not presented a significant discussion of the limitations of the proposed approach. For example, under what conditions should this benchmark be relied upon to declare quantum supremacy (or not)?

**Opportunities For Improvement:**

(-) What is the Y-axis of Figure 1?
(-) Table 4 needs to be discussed further. As the number of qubits goes from 100 to 500, the speedup compared to CTG-Kaypar [13] falls from about 10^4 to about 10^2. It would be interesting to extend the table and check when the simulation is no longer feasible or cannot do better than [13]. A discussion of the underlying reasons for this relative slowdown should also be discussed.
(-) The section moving from TNCO using Ising model to TNCO using K-spin Ising model is only described in a single sentence. How does the memory size of the model change with an increase in the value of K? Also, what is the value of K for Tables 1 through 4.

**Relation To Prior Work:**

Prior work is adequately described and they are also included in the submitted baselines.

**Summary And Contributions:**

The paper presents a Hamiltonian-based reinforcement learning approach for efficiently simulating quantum circuits by posing the same as a tensor network contraction ordering problem and modeling it using the K-spin Ising model. A dozen massively parallel environments to simulate quantum circuits have been developed.

---

> ### Author Response · Authors · 2023-08-28
>
> **Q1: What is the Y-axis of Figure 1?**
>
> __A1:__ The Y-axis is time, which is the estimated simulation time on electrical computers. We added the label to Y-axis. However, due to the large range of classical simulation time by different methods, it is not easy to add the ticks to the Y-axis. For Google's Sycamore circuits with 53 qubits, regarding the random number generation task, Google's original paper [1] reported an  estimated simulation time of 10,000 years.  The previous  work [2] reported an estimation of 21 days with 53 qubits. In this paper, we further reduce the simulation time to less than **4** days.
>
> [1] Arute, Frank, et al. "Quantum supremacy using a programmable superconducting processor." Nature 574.7779 (2019): 505-510.
> [2] Huang, Cupjin, et al. "Classical simulation of quantum supremacy circuits." arXiv preprint arXiv:2005.06787 (2020).
>
>
>
> **Q2:  Table 4 needs to be discussed further. As the number of qubits goes from 100 to 500, the speedup compared to CTG-Kaypar [13] falls from about 10^4 to about 10^2. It would be interesting to extend the table and check when the simulation is no longer feasible or cannot do better than [13]. A discussion of the underlying reasons for this relative slowdown should also be discussed.**
>
> __A2:__ For the random number generation problem, the computation complexity grows exponentially with increasing the number of qubits. Despite the utilization of our RL-based method to search the  optimal contraction path, the fundamental complexity of the problem results in an inherent exponential increase of simulation time. Consequently, this exponential growth inherently curtails the speedup in comparison to CTG-Kahypar. We believe it is possible to further improve the speedup in larger-scale case, with the help from the open-source community.
>
> The research on quantum supremacy aims to find a first demonstration that a programmable quantum device can solve a problem with polynomial time, while no classical computer can solve in any feasible amount of time.  We agree it is meaningful to  extend the Table 4, identify the limit of our method, and give an unequivocal first demonstration for the "quantum supremacy".
>
>
>
>
>
>
>
> **Q3: The section moving from TNCO using Ising model to TNCO using K-spin Ising model is only described in a single sentence. How does the memory size of the model change with an increase in the value of K? Also, what is the value of K for Tables 1 through 4.**
>
>
> __A3:__ For $N$ tensor network contraction environments, we independently sample trajectories with length $L$ of the tensor network contraction order with the corresponding  numbers of multiplications. We store the sampled trajectories in the buffer, and fetch $B$ samples for training every  iteration. K-spin Ising model employs $K$ steps of the sampled trajectories for the computation of Hamiltonian term. The memory sizes for different value of K are same.
>
> We make our experiments on a server which has two Intel(R) Xeon(R) Gold 5118 CPUs. Each of CPUs has 12 cores @2.30GHz supporting 24 hardware threads. There are 8 NVIDIA A100 GPUs which consists of 48 GB device memory. There are 128 GB DDR4 memories on the server. We set the batch size  $B=32$, the number of parallel environments $N=1024$, the length of trajectories $L=20$,  $K=3$.
>
>
> **Q4: The paper has not presented a significant discussion of the limitations of the proposed approach. For example, under what conditions should this benchmark be relied upon to declare quantum supremacy (or not)?**
>
> __A4:__ One limitation of this paper is our focus on the superconducting quantum circuits, i.e., Google's Sycamore circuits. There are some other types of quantum circuits, such as the photonic quantum computer [1] and Ising machine [2].
>
>
> [1] Oszmaniec, Michał, and Daniel J. Brod. "Classical simulation of photonic linear optics with lost particles." New Journal of Physics 20.9 (2018): 092002.
> [2] Kraus, Barbara. "Compressed quantum simulation of the ising model." Physical review letters 107.25 (2011): 250503.

---

> > ### Comment · Reviewer_YFU3 · 2023-08-31
> >
> > > due to the large range of classical simulation time by different methods, it is not easy to add the ticks to the Y-axis.
> >
> > A solution to this is to use a logarithmic scale for the Y-axis

---

### Official Review · Reviewer_K8x3 · 2023-07-24

**Rating:** 6
**Confidence:** 3
**Correctness:** Some variational circuit based method…
**Clarity:** The section 1 needs to be improved.

**Strengths:**

Overall the code quality is reasonable. Although some connection to exiting compatible connection to `cuQuantum`, `Qiskit`, `Pennylane` are missing

**Additional Feedback:**

None.

**Documentation:**

It would be a plus by having real quantum device loading considering error correction.

**Ethics:**

None.

**Limitations:**

In its current state, I would rate this paper between a 4 to 6. However, given the issues with writing quality and the significant gaps concerning Variational Quantum Eigensolver (VQE) and Variational Quantum Classifier (VQC) baselines, the draft requires substantial revisions. For instance, Section 1's walkthrough should highlight different quantum simulation-based RL solutions and more precisely define RL in the title to reflect the actual benchmarking efforts.

**Opportunities For Improvement:**

Additionally, the sole reliance on GPU-based simulation benchmarks could be viewed as a drawback. In the field of quantum computing, this is not entirely empirical due to the need for error correction.

A potential avenue for improvement could be the incorporation of device noise loading [1] during quantum simulation. The authors may want to consider this suggestion to enhance the next iteration of their work or any future submissions.

**Relation To Prior Work:**

Needs to be very improved.

**Summary And Contributions:**

This paper introduces a simulation codebase that utilizes a tensor train network. The primary innovation comes from leveraging the opt_einsum library to facilitate parallel computation within the tensor-train network.

The authors apply their approach within the context of reinforcement learning. However, a notable shortcoming is the paper's disproportionate emphasis on Google's claim of quantum supremacy. While this topic has been extensively debated within the quantum computing and quantum machine learning communities since 2019, the paper does not offer a robust analysis or evaluation of quantum advantage.

This paper would seem to overlook important contributions from variational circuit-based reinforcement learning (RL) work. This omission significantly undermines the comprehensiveness of the presented codebase.




- Confidence

The reviewer is confident about this review when giving tutorial in this sub-field since 2021.

***
**References**

1. Understanding and compensating for noise on IBM quantum computers
https://qiskit.org/documentation/stable/0.28/stubs/qiskit.providers.aer.noise.NoiseModel.html

2. "Theoretical error performance analysis for variational quantum circuit based functional regression." npj Quantum Information 9.1 (2023): 4.

3. Advances in quantum reinforcement learning." In 2017 IEEE International Conference on Systems, Man, and Cybernetics (SMC), pp. 282-287. IEEE, 2017.

4. QTN-VQC: An End-to-End Learning Framework for Quantum Neural Networks, Tensor-Network Workshop Neurips 2021

---

> ### Author Response · Authors · 2023-08-21
>
> **Q1: However, a notable shortcoming is the paper's disproportionate emphasis on Google's claim of quantum supremacy. While this topic has been extensively debated within the quantum computing and quantum machine learning communities since 2019, the paper does not offer a robust analysis or evaluation of quantum advantage.**
>
> __A1__: Yes, indeed, this paper targets at the classical simulation task of Google's quantum supremacy circuits. Our conclusion is that the search of "quantum supremacy" still __lacks an unequivocal first demonstration__. In this project, we propose to leverage reinforcement learning methods to establish benchmark performance of the classical simulation of quantum supremacy circuits. Open-source dataset and benchmark of the classical simulation tasks, including the random generation task in this project, will help avoid hasty advertisements of "quantum supremacy" [1] in the future.
>
> For the random number generation task, Google's "quantum supremacy [1]" announcement relied on an wrongly estimated simulation time of 10 000 years on the Summit supercomputer. In our updated version, we further reduce it to less than __4__ days. That empirical claim of quantum supremacy by Google team [3] should have been rejected by academia, and such a "quantum supremacy" claim shall be viewed as Google's hasty advertisement.
>
>
> The research on quantum advantage is about quantum mechanisms that provide theoretical speedup over conventional algorithms [2], e.g., $O(\sqrt N)$ for Bayesian inference, $O(\log N)$ for least-squares fitting, etc. The research on quantum supremacy aims to find a first demonstration that a programmable quantum device can solve a problem with  polynomial time,  while no classical computer can solve in any feasible amount of time, irrespective of the usefulness of the problem [3].
>
>
>
> [1] Arute, Frank, et al. "Quantum supremacy using a programmable superconducting processor." Nature 574.7779 (2019): 505-510.
> [2] Biamonte, Jacob, et al. "Quantum machine learning." Nature 549.7671 (2017): 195-202.
> [3] Lund, Austin P., Michael J. Bremner, and Timothy C. Ralph. "Quantum sampling problems, BosonSampling and quantum supremacy." npj Quantum Information 3.1 (2017): 15.
>
> **Q2: This paper would seem to overlook important contributions from variational circuit-based reinforcement learning (RL) work. This omission significantly undermines the comprehensiveness of the presented codebase.**
>
> __A2__: In this paper, we work on the classical simulation of quantum supremacy circuits. The task is mathematically the tensor network contraction problem by mapping the quantum circuits into a tensor network. We use a classical RL algorithm to search for the optimal contracting ordering, which provides the most efficient simulation of quantum circuits on classical computers. This research can be viewed as using clasical algorithm to address a quantum task. We do not consider the variational circuit-based RL methods (namely, quantum RL algorithms).

---

> > ### Author Response · Authors · 2023-08-21
> >
> > **Q3: Additionally, the sole reliance on GPU-based simulation benchmarks could be viewed as a drawback. In the field of quantum computing, this is not entirely empirical due to the need for error correction. A potential avenue for improvement could be the incorporation of device noise loading [1] during quantum simulation. The authors may want to consider this suggestion to enhance the next iteration of their work or any future submissions.**
> >
> > __A3__: There may be a misunderstanding regarding the usage of GPUs. Please note that GPUs were NOT used for simulating quantum circuits. The classical simulation task (which aims to find the best performance on an electric computer) of noisy Supremacy quantum circuits is mathematically a tensor network contraction task (searching for the optimal contracting ordering), as given in Fig. 3~4 with description in Section 3. Here, GPUs were used to sample training data that are feed into a policy network, since we used an RL algorithm. GPU provides massive parallel Monte Carlo simulation (of the searchig process on electric computers), speeding up the data sampling process about $1,000 \times$. Therefore, our GPU-based implementation is a substantial advantage for the proposed RL algorithm.
> >
> > We agree that the error correction is ensential for quantum computig due to device noise. The Sycamore quantum supermacy circuits have taken device noise into  account. The fedality of generated random numbers is measured by the XEB cross-entropy metric [1].  The Google team [1] used the following noise model for the random numbers produced by the Sycamore circuits,
> > $$ N_c(x) = \phi P_C + (1-\phi)2^{-n},$$
> > where $P_{C}$ is the probability distribution of a perfect quantum circuits and $\phi$ is a parameter measuring the reliability since the quantum circuits are noisy. Note that $\phi$ was estimated [1] as follows,
> > $$\phi=\prod_{g \in G_{1}}(1-e_{g_1})\prod_{g \in G_{2}}(1-e_{g_2})\prod_{q \in Q_{1}}(1-e_q),$$
> >
> > where  $G_{1}$ is the set of single-qubit gate, $G_{2}$ is the set of  double-qubits gate, and $Q_{1}$ is the set of qubits. Google set [1]  $e_{g_1}=0.16\%$, $e_{g_2}=0.62\%$ and $e_{q}=3.8\%$.
> > The quality of generated random numbers is measured by the XEB cross-entropy metric as follows,
> > $$ F_{XEB}(\hat{x})=\frac{1}{N}\sum_{i=1}^{N}2^n P_{C}(\hat{x}^{(i)})-1,$$
> >
> > The target task, classical simulation of quantum supremacy circuits, aims to provide efficient simulations of $P_C$ on electronic computers. As mentioned inthe above, it is a tensor network contraction task, as given in Fig. 3~4 with description in Section 3.
> >
> >
> >
> > In order to make the above discussions clear to readers, in Appendix A we added description of noise model, XEB cross-entropy metric, parameter $\phi$.
> >
> > [1] Arute, Frank, et al. "Quantum supremacy using a programmable superconducting processor." Nature 574.7779 (2019): 505-510.
> >
> > **Q4: In its current state, I would rate this paper between a 4 to 6. However, given the issues with writing quality and the significant gaps concerning Variational Quantum Eigensolver (VQE) and Variational Quantum Classifier (VQC) baselines, the draft requires substantial revisions. For instance, Section 1's walkthrough should highlight different quantum simulation-based RL solutions and more precisely define RL in the title to reflect the actual benchmarking efforts.**
> >
> >
> > __A4__: The target task is the classical simulation of quantum circuits. For the random number generation task, Google’s “quantum supremacy” announcement [1] relied on an estimated simulation time of 10 000 years on the Summit supercomputer.  We map the quantum circuits to the tensor network and transform the classical simulation of quantum circuits into the tensor network contraction problem. Our solution uses RL to search the optimal tensor network contraction path, which involves the minimum number of multiplications. We reduce the simulation time to less than __4__ days, which implies that the "quantum supremacy" has not been achieved with the Sycamore circuits at the scale of 53 qubits.
> >
> > The VQE [2] and VQC [3] are variational circuit-based quantum algorithms, which are different from the classical simulation of quantum circuits and not used for the announcement of "quantum supremacy".
> >
> >
> >
> > [1] Arute, Frank, et al. "Quantum supremacy using a programmable superconducting processor." Nature 574.7779 (2019): 505-510.
> > [2] Tilly, Jules, et al. "The variational quantum eigensolver: a review of methods and best practices." Physics Reports 986 (2022): 1-128.
> > [3] Schuld, Maria, et al. "Circuit-centric quantum classifiers." Physical Review A 101.3 (2020): 032308.

---

> > > ### Author Response · Authors · 2023-08-21
> > >
> > > **Q5: Some variational circuit based method is missing.**
> > >
> > > __A5__: In this paper, we challenge the announcement of quantum supremacy. It relies on the wrongly simulation time on the Summit supercomputer for the random number generation task. We use a classical RL algorithm to accelerate the simulation process. Our approach is not related to the variational circuit-based RL methods.
> > >
> > > **Q6: The section 1 needs to be improved.**
> > >
> > > __A6__: Thanks for your suggestion. We make the following revisions.
> > >
> > > 1. We missed the notation of the Y-axis in Fig. 1 of the Section.1. It is the running time of different classical quantum simulation.  We added the label to the Y-axis.
> > > 2. We gave a clear conclusion of our work, that the "quantum supremacy" claim still lacks an unequivocal first demonstration.
> > >
> > > **Q7: It would be a plus by having real quantum device loading considering error correction.**
> > >
> > > __A7__: Thanks for your suggestion. This project has three stages: searching for the optimal tensor network contraction path, evaluating the real running time, and evaluating the quality of generated random numbers. In this paper, we focus on the first stage, identifying the optimal tensor network contraction path, to accelerate the classical simulation of quantum circuits. Future work remains to have real quantum device loadings to evaluate generated random numbers.

---

### Author Response · Authors · 2023-08-28
**Summary and Thanks to All Reviewers and Area Chair**

The authors sincerely thank all reviewers and area chair. To recap, this work has made the following major contributions.

1. We use the machine learning to establish the best results for the classical simulation of quantum supremacy circuits. In particular, our reinforcement learning approach demonstrate a great potential by reporting an estimated simulation time of __less than 4 days__, a speedup of 4.62× over the state-of-the-art method.

3.  We develop a dozen of __massively parallel GPU-based implementations__ of the quantum circuits represented by tensor network contraction environments, including tensor-train, synthetic, and sycamore quantum circuits.

4. We provide various __benchmark__ methods, including the OE-greedy, CTG-Greedy, CTG-Kahypar, and RL-TNCO, and design __a unified representation__ of the tensor network contraction order found by different methods. We establish fair comparison for future follow-up works.

__Our conclusion__ is that the search of “quantum supremacy” still lacks an unequivocal first demonstration. The long-term collaborations from the AI/ML and quantum physics communities are required to maintain reference curves for validating the “empirical quantum supremacy” and drive continuing hardware advancements.

---

### Decision · Program_Chairs · 2023-09-22

**Decision:**

Accept (Poster)

**Comment:**

The work sets out to challenge the "quantum supremacy" claim put forward by Google in 2019 (NB: the claim has been found to be bogus before by others), demonstrating that the by Google projected 10K years claim to reproduce the results classically, can essentially be done in 4 days, essentially rejecting the supremacy claim. To avoid future vendor-driven advertisement campaigns of wrongful nature, the authors publish code and data (i.e., scripts to produce data) that would help to check future circuits against supremacy claims to avoid misleading statements of the nature reported by Google in Nature!

A fruitful discussion among authors and reviewers led to improvements being applied to the paper throughout the process.

It is our impression that there still remains some confusion as to the data provided by the authors, i.e., binary vs. text; whether the data is provided at all or only scripts to generate the data. We'd like to encourage the authors to make these aspects crystal clear and if in doubt provide samples (or the entire data) of the data generated by the scripts such that others would be able to reproduce the results without much guess work involved. Thank you.